# The Association between Maternal Stress and Glucocorticoid Rhythmicity in Human Milk

**DOI:** 10.3390/nu13051608

**Published:** 2021-05-11

**Authors:** Michelle Romijn, Luca J. L. van Tilburg, Jonneke J. Hollanders, Bibian van der Voorn, Paul de Goede, Koert M. Dolman, Annemieke C. Heijboer, Birit F. P. Broekman, Joost Rotteveel, Martijn J. J. Finken

**Affiliations:** 1Department of Pediatric Endocrinology, Emma Children’s Hospital, Amsterdam University Medical Centers, location VUmc, 1081 HV Amsterdam, The Netherlands; l.vantilburg@amsterdamumc.nl (L.J.L.v.T.); j.hollanders@amsterdamumc.nl (J.J.H.); j.rotteveel@amsterdamumc.nl (J.R.); m.finken@amsterdamumc.nl (M.J.J.F.); 2Department of Pediatrics, Amsterdam Reproduction & Development Research Institute, Vrije Universiteit Amsterdam, Amsterdam UMC, 1081 HV Amsterdam, The Netherlands; 3Department of Pediatrics, Division of Endocrinology, Erasmus MC-Sophia Children’s Hospital, University Medical Center Rotterdam, 3000 CA Rotterdam, The Netherlands; b.vandervoorn@erasmusmc.nl; 4Laboratory of Endocrinology, Amsterdam University Medical Center, University of Amsterdam, Amsterdam Gastroenterology & Metabolism, 1105 AZ Amsterdam, The Netherlands; pauldegoede@gmail.com; 5Department of Pediatrics, Onze Lieve Vrouwe Gasthuis (OLVG), 1006 AE Amsterdam, The Netherlands; k.dolman@olvg.nl; 6Department of Clinical Chemistry, Endocrine Laboratory, Vrije Universiteit Amsterdam, Amsterdam UMC, 1081 HV Amsterdam, The Netherlands; a.heijboer@amsterdamumc.nl; 7Department of Psychiatry, VU University Medical Centre, Amsterdam UMC, 1081 HV Amsterdam, The Netherlands; B.F.P.Broekman@olvg.nl

**Keywords:** hypothalamus–pituitary–adrenal axis, glucocorticoid rhythm, psychopathology, breast milk

## Abstract

Background: Chronic stress is often accompanied by alterations in the diurnal rhythm of hypothalamus–pituitary–adrenal activity. However, there are limited data on the diurnal rhythmicity of breast milk glucocorticoids (GCs) among women with psychological distress. We compared mothers who sought consultation at an expertise center for pregnant women with an increased risk of psychological distress with control mothers for GC diurnal rhythmicity in milk and saliva obtained at the same time. Methods: We included 19 mothers who sought consultation at the psychiatry–obstetric–pediatric (POP) outpatient clinic and 44 control mothers. One month postpartum, mothers collected on average eight paired milk and saliva samples during a 24 h period. GC levels were measured using liquid chromatography–tandem mass spectrometry. GC rhythmicity parameters were determined with specialized software. Results: For both milk and saliva, no group differences regarding GC rhythms were found. Milk cortisol area under the curve with respect to the ground was lower in the POP group than in the control group (*p* = 0.02). GC levels in human milk and saliva were highly correlated within each group (*p* < 0.001). Conclusion: Although there were no differences between groups in GC rhythmicity, the total amount of milk cortisol was lower in the POP group. Long-term follow-up is needed to address the impact of vertical transmission of breast milk GCs.

## 1. Introduction

The health benefits of breastfeeding for mother and child are acknowledged worldwide, and, therefore, breastfeeding is recommended as the feeding method of choice by the World Health Organization (WHO) [1,2]. Human milk provides nutrients and immune factors for infants that benefit neurodevelopment, growth, cardiovascular health and immunity [3]. The glucocorticoids (GCs) cortisol and cortisone are also present in human milk [4]. The production of these hormones is controlled by the hypothalamus–pituitary–adrenal (HPA) axis, and follows a diurnal rhythm, with a peak in the early morning and a nadir in the late evening [5,6]. Previous studies, including one from our group, have shown that GC levels in human milk follow a typical diurnal pattern, mirroring GC levels in saliva [7,8]. There is some evidence suggesting that GCs through human milk may be important for metabolism, neurodevelopment and immunity [9], although the evidence is not unequivocal [7,10].

The GC composition in human milk may be influenced by various factors, including maternal stress, weight, educational status and parity [9,11,12]. Studies addressing the relation between maternal stress and GC levels in human milk are scarce [9]. Chronic maternal stress, as seen in depressive and anxiety disorders, may be accompanied by changes in HPA axis activity, although the direction of association is not clear [13,14,15,16,17,18]. Explanations for these varying results can be found in differences across studies regarding the definition of maternal stress and the choice of the sampling protocol, including the number of samples collected. After all, to obtain a reliable estimate of the diurnal rhythm of the HPA axis, multiple measurements across a 24 h period are required.

The aim of this study was to compare GC rhythmicity in human milk collected during a 24 h period of mothers who sought consultation at the psychiatric–obstetric–pediatric (POP) outpatient clinic with that of control mothers. In addition, we aimed to compare GC levels in human milk with GC levels in saliva obtained at the same time.

## 2. Materials and Methods

### 2.1. Study Design and Participants

This study was part of the Cortisol in Mother’s Milk Study (COSMOS), for which mothers were recruited between March 2016 and July 2017. The COSMOS study is a prospective cohort study, which aimed to research the associations between GC rhythmicity in human milk and various infant outcomes [7,8,10].

For this study, mothers were recruited from the psychiatric–obstetric–pediatric (POP) outpatient clinic of the Onze Lieve Vrouwe Gasthuis (OLVG) in Amsterdam, the Netherlands (POP group). The POP outpatient clinic is an expertise center for women with an increased risk of psychological distress during pregnancy, including women with a history of psychiatric disease, clinically relevant symptoms of distress during pregnancy, or present use of psychotropic medication. Control mothers were recruited postpartum from the maternity ward of the Vrije Universiteit Medical Center (VUmc) in Amsterdam, the Netherlands (control group). These mothers were admitted for obstetric reasons (e.g., breech position or cesarean section). The Medical Ethics Committee (METC) of the VU University Medical Center approved the study (protocol number 2015.524), and written informed consent was obtained from all participants.

Mothers were eligible to participate in the study when they met the following inclusion criteria: birth at term (gestational age 37–42 weeks); an appropriate birth weight for gestational age (defined as a birth weight between −2 and +2 SD score); and, the intention to breastfeed their child for at least three months. Mothers with pregnancy-related disorders, such as preeclampsia or HELLP syndrome, multiple pregnancy, major congenital anomalies of their infant and/or alcohol use of >7 international units (IU) per week were not eligible for participation. In addition, during the study mothers were excluded in case of fever >38.5 °C (as measured by themselves) at the time of sampling.

### 2.2. Data Collection

During the first week postpartum, mothers filled in a questionnaire regarding their pregnancy, delivery, anthropometric and demographic data.

One month (±5 days) postpartum, mothers collected 1–2 mL human milk before every feeding moment (on average 8 times a day) during a 24 h period. Milk extraction was performed manually or electrically, dependent on the preference of the mother, as long as they performed it the same way for each collection moment. Simultaneously, maternal saliva was collected using a Salivette cotton swab (Sarstedt). The cotton swab had to be placed in the mouth for about 60 s, and in order to stimulate salivation, participants had to chew on the swab. After collection, the milk and saliva samples were stored in the freezer at home, until the samples were collected by the research team within approximately one week.

In addition, information on self-reported levels of depressive and anxiety symptoms was collected one month postpartum with the use of the self-reported Hospital Anxiety and Depression Scale (HADS) questionnaire. This questionnaire consists of 7 questions regarding depressive symptoms (hospital depression subscale; HADS-D), and 7 questions regarding anxiety symptoms (hospital anxiety subscale; HADS-A), with a scoring range from 0 to 3 for each item. A score of 8 points or more on 1 of the 2 subscales HADS-D or HADS-A, is considered as clinically relevant depressive and/or anxiety symptoms [19].

### 2.3. Laboratory Analysis

At the VUmc, milk and saliva samples were stored at −20 °C for less than 3 months, and thawed only once prior to analysis. GC levels in human milk and saliva were determined by using isotope dilution liquid chromatography–tandem mass spectrometry (LC–MS/MS), as described previously [20]. In short, lipids were removed by hexane washing the human milk samples three times, after adding internal standards (13C3-labeled cortisol and 13C3-labeled cortisone). This procedure was not necessary for the saliva samples. Thereafter, samples were extracted using Isolate plates (Biotage, Uppsala, Sweden), and analyzed by using LC–MS/MS (Acquity with Quattro Premier XE, Milford, Water Corporation, MA, USA). For LC–MS/MS measurements, the intra-assay coefficients of variation were 4% and 5% for cortisol levels of 7 and 23 nmol/L, and 5% for cortisone levels of 8 and 33 nmol/L. The inter-assay coefficient of variation was <9%, and the lower limit of quantitation was, for both cortisol and cortisone, 0.5 nmol/L.

### 2.4. Statistical Analysis

#### 2.4.1. Glucocorticoid Rhythmicity Analysis

Rhythmicity of GC levels in human milk and saliva was analyzed using Gaussian peak regression with SigmaPlot 14.0 software (Systat Software, San Jose, CA, USA) for both groups separately. Data were tested for single- and double-peak appearance, and eventually the data were best fitted to the following regression formula: y = a*exp(–0.5*((x–x0)/b)^2), where ‘a’ represents the estimate for the peak height, ‘b’ represents the estimate for the peak width and ‘x0′ represents the estimate for the peak timing across the 24 h period. GC rhythmicity in human milk and saliva was compared between the POP group and the control group by using Graphpad Prism 8.2.1.

#### 2.4.2. Other Analyses

The area under the curve with respect to the ground (AUCg) was calculated using the trapezoid rule [21]. The AUCg in human milk represents the infant GC exposure. Because collection times differed between the mothers, AUCg values were corrected for the time and the duration of collection. AUCg/h values were compared between groups by performing a Mann–Whitney U test.

Correlations between GC levels in human milk and saliva for the POP group and the control group separately were analyzed by using generalized estimating equations (GEE) analysis, which allows correction for multiple measurements within participants. The data were transformed to Z-scores to provide correlation coefficients. These coefficients are expressed as r’s with 95% confidence intervals. In addition, the correlations between the groups were investigated by performing GEE analysis including an interaction term for each study group. In order to investigate whether the correlations were influenced by HADS score, socioeconomic status (*Z*-score based on average income, % low income, % low-skilled and % unemployed civilians per postal code area, with data from the Dutch Social Cultural Project office (2014, the Netherlands)) and use of antidepressants, adjusted GEE analysis was performed. The groups did not differ in BMI and parity (see Table 1), and these factors were kept out of the analyses. A *p*-value of <0.05 was considered statistically significant.

## 3. Results

### 3.1. Patient Characteristics

A total of 236 mothers were approached for this study, of which 91 agreed to participate. Twenty-eight mothers were lost to follow-up or withdrew consent, leaving 63 mothers with complete data. Among them were 19 mothers who sought consultation at the POP clinic and 44 control mothers (Figure 1).

The characteristics of participants are shown in Table 1. The POP group had a higher HADS score postpartum, and more often reported the use of antidepressants than the control group.

### 3.2. Glucocorticoid Rhythmicity

Figure 2A–D show the GC levels across the diurnal cycle for both human milk and maternal saliva by group. In both groups, a typical diurnal rhythm is visible, with an increase in the morning and a subsequent decline of GC levels.

In both groups a monophasic cortisol and cortisone rhythm was detectable in both media (*p* < 0.001 for overall fit and peak placement). Peak height, width and timing did not differ between the groups (Table 2).

### 3.3. Glucocorticoid Levels

In Table 3 the AUCg values are shown for both cortisol and cortisone levels in human milk and saliva. The AUCg for cortisol in human milk was higher in the control group compared to the POP group (*p* = 0.02). The AUCg for cortisol in saliva and the AUCg for cortisone in both human milk and saliva were not different between the groups, although the direction of association was similar.

Table 4 presents the results of the GEE analysis, and shows that GC levels in human milk and saliva were strongly correlated within each group. The correlations were not different between the groups. Adjusted GEE analysis showed that correction for HADS score, SES and use of antidepressants did not influence these correlations.

## 4. Discussion

The main finding of our study was that GC levels in human milk obtained at one month postpartum from mothers seeking consultation at the POP clinic showed a clear monophasic rhythm, strongly correlated to the GC rhythm in maternal saliva, which did not differ from control mothers. We found that, in spite of similar rhythm parameters, the AUCg for milk cortisol was lower among mothers from the POP group than among control mothers independent of the number of available samples.

Findings from previous studies investigating associations between experienced stress and GC level or rhythmicity in various samples [22], including serum [14,23], saliva [13,16,24], (early morning, spot or 24 h) urine [15], hair [25] and milk [12,26,27], were contradictory. There are several explanations for these different findings. First, studies did not collect enough samples for the study of HPA axis rhythmicity. In the majority of previous studies, only one or two samples per diurnal cycle were collected. In addition, not only the number of samples is important, but also the timing of sampling. One study suggested that women with increased psychological stress collected their morning samples approximately three hours after awakening versus half an hour post-awakening in women with low stress experience [12]. Since GCs reach their highest value approximately 30–45 min after awakening, it might be possible that the lack of association could be attributed to between-group differences in sampling times [28]. However, this seemed not to be the case in our study, since both groups donated approximately the same number of samples per 2 h time interval. Second, studies differed in the duration of stress exposure. Acute stress triggers the HPA axis to become active, leading to an acute elevation of GC levels. Chronic stress may be accompanied by exhaustion of the HPA axis, characterized by a blunted diurnal rhythm and a decreased responsiveness to stressors [17]. It is possible that the women in the POP group had evidence of chronic stress exposure. Although there were no differences between groups in GC rhythmicity, the POP group had a lower AUCg for milk cortisol, indicative of a lower infants’ GC exposure. This is in line with a previous study, demonstrating that newborns born to mothers who sought consultation at the POP clinic had a lower cortisol concentration in hair [25].

The absence of major differences between our study groups contrasts with findings from previous studies addressing the impact of depression and anxiety on GC rhythms in women [17,29,30,31,32,33]. However, mothers seeking consultation at the POP clinic in our study received tailored care, as was suggested by the relatively low HADS scores postpartum. In addition, the majority of women seeking consultation at the POP clinic used antidepressants, which may alter the diurnal cortisol pattern [34,35]. An alternative explanation for the similar findings between the study groups is that mothers on the maternity ward may have experienced higher stress exposure, including the higher number of cesarean sections performed in this group, although it is unclear whether this persists beyond admission.

Another finding of our study was that GCs in human milk and maternal saliva were highly correlated, which is in line with our previous study in a smaller sample [8]. It has been argued that circulating GCs enter the mammary gland by passive diffusion across a concentration gradient [4]. As such, future studies in this field might consider collecting only saliva samples, as a proxy for milk GC levels. The observation that the strength of the correlations did not differ between the study groups strongly suggests that maternal stress does not seem to influence any form of active transport.

This study has multiple strengths. First of all, the mothers collected on average eight samples during a 24 h period, which allowed us to reliably analyze GC rhythms throughout the day. Second, to the best of our knowledge this is the first study comparing GC levels across the full diurnal cycle between human milk and saliva in mothers seeking consultation at the POP clinic and control mothers. Third, in addition to GC measurement in human milk and saliva, the mothers filled in self-perceived stress scores which made it possible to compare biological stress parameters with the subjectively perceived stress parameters.

This study also has its limitations. First, the sample size of our study population, in particular the POP group, was limited. Second, apparently mothers seeking consultation at the POP clinic received effective treatment, as was evidenced by the relatively low HADS scores, which could have decreased the differences between the POP group and the control group. However, this is not a limitation in clinical practice. Concurrently, the mothers in the control group were not representative of the general population, since they were admitted for medical reasons that may be accompanied by a higher stress level. However, for our study it is reasonable to assume that stress levels experienced at birth would have normalized by the time of sampling. Third, unfortunately our study lacked information regarding maternal sleeping behavior, which is known for a strong association with HPA axis rhythmicity. Lastly, the sample collection was restricted to a single day at one month postpartum, and day-to-day variability in HPA axis activity could therefore not be taken into account.

## 5. Conclusions

GC levels in human milk obtained at one month postpartum from mothers seeking consultation at the POP clinic showed a clear monophasic rhythm and were not different across the full diurnal cycle from milk GC levels of control mothers. However, despite no differences in GC rhythmicity, we found that the total cortisol exposure was lower in human milk from mothers from the POP clinic compared to healthy mothers. Long-term follow-up is needed to address the impact of vertical transmission of breast milk GCs.

## Figures and Tables

**Figure 1 nutrients-13-01608-f001:**
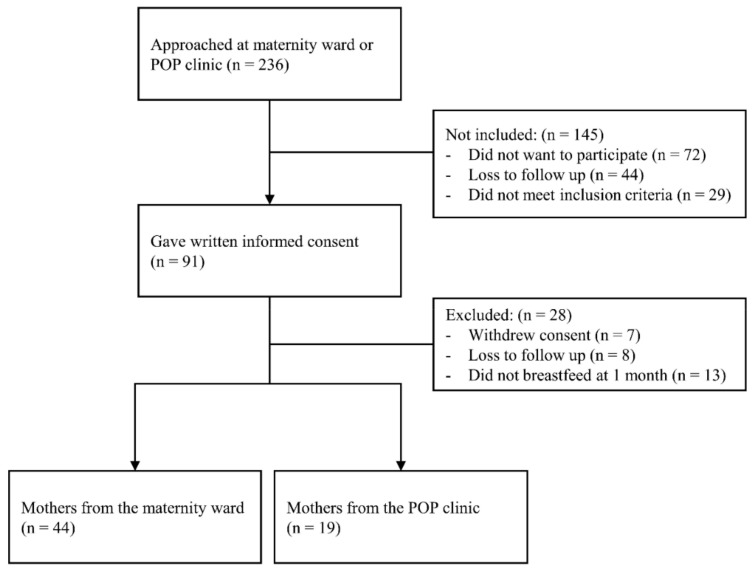
Flowchart of the inclusion process. POP clinic = psychiatric-obstetric-pediatric clinic.

**Figure 2 nutrients-13-01608-f002:**
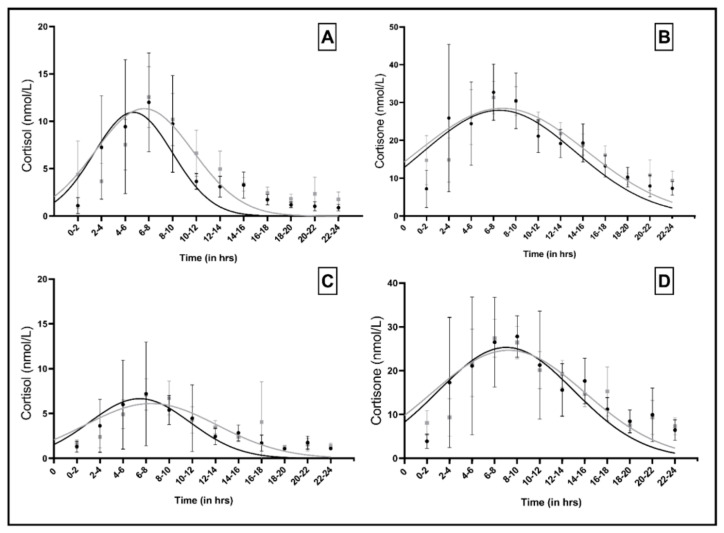
GC rhythms in human milk (**A**,**B**) and saliva (**C**,**D**) of the POP group (black dots and black lines) and the control group (grey squares and grey lines).

**Table 1 nutrients-13-01608-t001:** Characteristics of the study population by group.

	Total *N* = 63	POP Group *N* = 19 (30%)	Control Group *N* = 44 (70%)
Mothers’ age, years—mean (±SD)	33.7 (±4.6)	32.5 (±4.6)	34.6 (±4.6)
Gestational age, weeks—mean (±SD)	39.7 (±1.3)	39.5 (±1.4)	39.8 (±1.4)
Parity—median [IQR]	2 [1,2]	2 [1,2]	1 [1,2]
BMI—mean (±SD)	22.46 (±2.6)	22.69 (±2.2)	22.36 (±2.8)
Birth weight, grams—mean (±SD)	3509 (±479)	3434 (±436)	3541 (±497)
Cesarean section—number (%)	26 (42%)	4 (21%)	22 (50%) *
Antidepressant medication—number (%)	14 (22%)	11 (58%)	2 (5%)
HADS score 1 month postpartum HADS-A Score—mean (±SD) HADS-D Score—mean (±SD)	4.10 (±3.10) 2.66 (±3.10)	5.58 (±3.69) 4.11 (±4.80)	3.44 (±2.59) * 2.02 (±1.67) *
Socioeconomic status—mean (±SD)	0.49 (±1.21)	0.41 (±1.24)	(±1.21)

* *p* < 0.05; POP = psychiatric-obstetric-pediatric; SD = standard deviation; HADS = Hospital Scale for Depression and Anxiety.

**Table 2 nutrients-13-01608-t002:** Peak height, width and timing of cortisol and cortisone in milk and saliva by group.

Outcome	POP Group	Control Group	*p*-Value
	Milk Cortisol	Milk Cortisol	
Peak height (nmol/L)	10.95 (±0.97)	11.34 (±0.69)	0.75
Peak width (h)	1.66 (±0.20)	2.06 (±0.17)	0.17
Peak timing (in 2 h)	3.34 (±0.20)	3.80 (±0.17)	0.12
	Milk Cortisone	Milk Cortisone	
Peak height (nmol/L)	27.95 (±1.40)	28.49 (±0.92)	0.75
Peak width (h)	3.41 (±0.29)	3.74 (±0.21)	0.37
Peak timing (2 h)	4.26 (±0.28)	4.40 (±0.19)	0.69
	Saliva Cortisol	Saliva Cortisol	
Peak height (nmol/L)	6.65 (±0.83)	6.11 (±0.50)	0.57
Peak width (h)	2.18 (±0.36)	2.87 (±0.36)	0.26
Peak timing (2 h)	3.72 (±0.38)	4.21 (±0.33)	0.39
	Saliva Cortisone	Saliva Cortisone	
Peak height (nmol/L)	25.31 (±1.95)	24.65 (±1.05)	0.75
Peak width (h)	3.02 (±0.35)	3.40 (±0.24)	0.380
Peak timing (2 h)	4.52 (±0.34	4.62 (±0.21)	0.80

Values represent mean (±SEM). Nmol/L = nanomol/liter; h = hours.

**Table 3 nutrients-13-01608-t003:** AUCg for cortisol and cortisone in human milk and saliva by group.

Variable	POP	Control
AUCg Cortisol Milk, median [IQR]	3.42 [2.63–4.59]	4.73 [3.59–6.10] *
AUCg Cortisone Milk, median [IQR]	16.12 [13.19–19.85]	17.86 [15.72–21.63]
AUCg Cortisol Saliva, median [IQR]	2.53 [1.78–2.92]	3.07 [2.32–3.92]
AUCg Cortisone Saliva, median [IQR]	13.49 [10.40–19.14]	15.34 [12.54–18.80]

AUCg = area under the curve with respect to the ground; IQR = interquartile range. * = *p* < 0.05.

**Table 4 nutrients-13-01608-t004:** Correlation (r) between GC levels in human milk and saliva within each group and between groups.

Glucocorticoid	Study Group	Correlation between Milk and Saliva (r)	95% CI
Cortisol	POP	0.841 *	0.683–1.000
	Control	0.829 *	0.702–0.956
	POP versus control	0.070	−0.140–0.279
Cortisone	POP	0.861 *	0.746–0.975
	Control	0.896 *	0.819–0.974
	POP versus control	−0.023	−0.162–0.107

‘r’ represents the correlation between milk and saliva measured with GEE analysis. 95% CI = 95% confidence interval. * = *p* < 0.001.

## Data Availability

The data presented in this study are available on request from the corresponding author. The data are not publicly available due to privacy and ethical restrictions.

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
