# Peer review of "The Association between Maternal Stress and Glucocorticoid Rhythmicity in Human Milk"

_nutrients, 2021, doi:10.3390/nu13051608_

Round 1

Reviewer 1 Report

The reviewer was pleased to review the work of Michelle Romijn et al. The reviewer did a comprehensive literature review about this similar topic. The reviewer found this work is good and with novelty. First, the author designed a good study to answer the unanswered question. The author designed a good protocol to investigate the circadian rhythms of GC and HM. The authors intensively collected one day HM from breastfed mother. The result did show the GC's circadian rhythms of HM. Second, the novelty is the establishment of the strong link of GC in saliva and GC in human milk. This finding may extend the future research of GC in human milk. Third, the authors chose a good population: postpartum stressed mothers. This is an important issue for the women and children health.

The reviewer has few suggestions.

  1. Please clarify the definition of socio-economic status with references.
  2. Control mothers were recruited postpartum from the maternity ward. Did the author have the diagnosis of the control group? The authors could detail "obstetric medical reasons".

Author Response

Reviewer 1
Comment reviewer 1: The reviewer was pleased to review the work of Michelle Romijn et al. The reviewer did a comprehensive literature review about this similar topic. The reviewer found this work is good and with novelty. First, the author designed a good study to answer the unanswered question. The author designed a good protocol to investigate the circadian rhythms of GC and HM. The authors intensively collected one day HM from breastfed mother. The result did show the GC's circadian rhythms of HM. Second, the novelty is the establishment of the strong link of GC in saliva and GC in human milk. This finding may extend the future research of GC in human milk. Third, the authors chose a good population: postpartum stressed mothers. This is an important issue for the women and children health.

The reviewer has few suggestions.

  1. Please clarify the definition of socio-economic status with references.
  2. Control mothers were recruited postpartum from the maternity ward. Did the author have the diagnosis of the control group? The authors could detail "obstetric medical reasons".

Reply: We would like to thank the reviewer for the evaluation and the nice compliments on our work. We also thank the reviewer for the valuable suggestions. The first suggestion regarding socio-economic status which is based on average income, % low income, % low-skilled and % enamplyed civilians per postal code area, with data from the Dutch Social Cultural Project [ref], and presented as a z-score. We have added this information in lines 153-155. The second suggestion regarding the diagnosis of the control group, we have decided to summarize the diagnosis of the control group as ‘obstetric medical reasons’ since these were all general obstetric reasons why admission at a hospital was necessary. For example, a caesarean section in history, breech position of wish for analgesia etc. In order to make this more clear we have changed this information into ‘These mothers were admitted for obstetric medical reasons (e.g. breech position or cesarean section)’ in line 87.

Reviewer 2 Report

The article aims to compare glucocorticoids rhythmicity in human milk and saliva collected during a 24-hour period of mothers who had consulted a psychiatric-obstetric-pediatric (POP) outpatient clinic before partum with that of control mothers. Mothers collected, one month postpartum, on average 8 paired milk and saliva samples during a 24-hour period. The results suggest that there is not any correlation between glucocorticoids (in milk and saliva) from POP group and the control group. Moreover, the results also show that the total amount of milk cortisol was higher in the control group. The relevance of the data is important, even though further studies are required because the sample size of the POP control is small, and samples are taken only during 1 day (correctly mentioned at the discussion). Some points need minor revisions before considering the manuscript suitable for publication.

This article presents an interesting and valuable discussion, I think that this manuscript is well-researched and an important contribution to knowledge.

General

Material and methods:

The authors could also mention the range of the mothers’ age.

In the 2.2 Data collection the authors should add the methods used for samples collection. How was milk extraction done? Was it a pump extraction, manual or electric?

Specific Comments:

  • Line 33: AUCg abbreviation should be mentioned first as the non-abbreviation form.
  • Line 49: Instead of “We and others have shown that GC levels in human milk follow a typical diurnal pattern, mirroring GC levels in saliva” we suggest using “Previous studies, including one from our group, have shown that GC levels in human milk follow a typical diurnal pattern, mirroring GC levels in saliva”.
  • Line 79: VU Medical center abbreviation should be mentioned first as the non-abbreviation form.
  • Figure 1 and figure 2 quality could be ameliorated.
  • All the abbreviation should be mentioned in the Tables description as the non-abbreviation form.
  • In lines 188 and 209 it can be found “GC levels” and “GCs levels” respectively, it should be written in a uniform manner.
  • Lines 194-196: “Findings from previous studies investigating associations between experienced stress and GC level or rhythmicity in various media, including serum, saliva, (early-morning, spot or 24-hour) urine , hair and milk, were contradictory[22] [13-16, 23-25] [12, 26, 27]”. The references should be next to what they are referencing. Moreover, instead of media I suggest the word samples.

Author Response

Reviewer 2

Comment reviewer 2: The article aims to compare glucocorticoids rhythmicity in human milk and saliva collected during a 24-hour period of mothers who had consulted a psychiatric-obstetric-pediatric (POP) outpatient clinic before partum with that of control mothers. Mothers collected, one month postpartum, on average 8 paired milk and saliva samples during a 24-hour period. The results suggest that there is not any correlation between glucocorticoids (in milk and saliva) from POP group and the control group. Moreover, the results also show that the total amount of milk cortisol was higher in the control group. The relevance of the data is important, even though further studies are required because the sample size of the POP control is small, and samples are taken only during 1 day (correctly mentioned at the discussion). Some points need minor revisions before considering the manuscript suitable for publication.

This article presents an interesting and valuable discussion, I think that this manuscript is well-researched and an important contribution to knowledge.

Reply: We would like to thank the reviewer for the evaluation of our work and for the valuable comments and suggestions.

General 

Material and methods: 

The authors could also mention the range of the mothers’ age.
Reply: We have added this information regarding the mean of mothers’ age in years in Table 1 with characteristics of the study population by group.

In the 2.2 Data collection the authors should add the methods used for samples collection. How was milk extraction done? Was it a pump extraction, manual or electric?
Reply: Thank you for this comment. We have added the information regarding method of milk extraction in the method section in lines 102-104: ‘Milk extraction was performed manually or electrical, dependent on the preference of the mother, as long as they performed it the same way for each collection moment’.

Specific Comments: 

  • Line 33: AUCg abbreviation should be mentioned first as the non-abbreviation form.

Reply: We have changed the abbreviation to the non-abbreviation form in the abstract in line 37.

  • Line 49: Instead of “We and othershave shown that GC levels in human milk follow a typical diurnal pattern, mirroring GC levels in saliva” we suggest using “Previous studies, including one from our group, have shown that GC levels in human milk follow a typical diurnal pattern, mirroring GC levels in saliva”.

Reply: Thank you for the suggestion. We have changed the sentence according to your suggestion (line 55).

  • Line 79: VUMedical center abbreviation should be mentioned first as the non-abbreviation form.

Reply: We have added the non-abbreviation form of VU (Vrije Universiteit) in line 85.

  • Figure 1 and figure 2 quality could be ameliorated.

Reply: The quality of figure 1 and figure 2 are ameliorated according to your suggestion.

  • All the abbreviation should be mentioned in the Tables description as the non-abbreviation form.

Reply: We have mentioned all the abbreviations in the Table descriptions according to your suggestion.

  • In lines 188 and 209 it can be found “GC levels” and “GCs levels” respectively, it should be written in a uniform manner.

Reply: We have corrected this typo into a uniform manner.

  • Lines 194-196: “Findings from previous studies investigating associations between experienced stress and GC level or rhythmicity in various media, including serum, saliva, (early-morning, spot or 24-hour) urine , hair and milk, were contradictory[22] [13-16, 23-25] [12, 26, 27]”. The references should be next to what they are referencing. Moreover, instead of media I suggest the word samples.

Reply: Thank you for this comment. We have placed the references next to what they are referencing and changed media into samples.
